# Calcium Intake in Children with Eczema and/or Food Allergy: A Prospective Cohort Study

**DOI:** 10.3390/nu11123039

**Published:** 2019-12-12

**Authors:** Hailey Hildebrand, Elinor Simons, Anita L. Kozyrskyj, Allan B. Becker, Jennifer L. P. Protudjer

**Affiliations:** 1Max Rady College of Medicine, The University of Manitoba, Winnipeg, MB R3E 0W2, Canada; hildeb87@myumanitoba.ca; 2Department of Pediatrics and Child Health, The University of Manitoba, Winnipeg, MB R3E 0W2, Canada; Elinor.Simons@umanitoba.ca (E.S.); Allan.Becker@umanitoba.ca (A.B.B.); 3The Children’s Health Research Institute of Manitoba, Winnipeg, MB R3E 3P4, Canada; 4Department of Pediatrics, The University of Alberta, Edmonton, AB T6G 1C9, Canada; kozyrsky@ualberta.ca; 5George and Fay Yee Centre for Healthcare Innovation, Winnipeg, MB R3E 0T6, Canada; 6Institute of Environmental Medicine, Karolinska Institutet, 171 77 Stockholm, Sweden

**Keywords:** adolescents, calcium, dairy, food allergy

## Abstract

Eczema and food allergy may impact diet. Using data from a cohort of Manitoba children born in 1995, we examined calcium intake, defined as the frequency and quality of calcium products consumed (with the exception of cheese), amongst Manitoba adolescents (12–14 years) with eczema or food allergy in childhood (7–8 years) or adolescence. At both ages, children were assessed by a physician for eczema and food allergy. Adolescents completed food frequency questionnaires. Calcium intake was defined as 1+ vs. <1 weekly. Linear and logistic regression was used as appropriate, with adjustments for confounders. Overall, 468 adolescents were included, of whom 62 (13.3%) had eczema only in childhood, 25 (5.3%) had food allergy only, and 26 (5.6%) had eczema and food allergy. Compared to children without eczema, those with eczema only had poorer calcium intake in adolescence (β −0.44; 95%CI −0.96; 0.00). Girls, but not boys, with eczema in childhood had poorer calcium intake in adolescence than girls without eczema (β −0.84; 95%CI −1.60; −0.08). These patterns persisted even if children experienced transient vs. persistent eczema to adolescence. Similar but non-significant trends were found for food allergy. Childhood eczema is associated with significantly lower calcium intake and consumption in adolescence. These differences persist to adolescence, even if a child “outgrows” their allergic condition.

## 1. Introduction

The allergic diseases, eczema and food allergy, now affect a large number of children. Globally, as many as 15%–30% of children live with eczema [1] and 4%–10% of children are directly affected by food allergy [2]. These diseases often involve dietary exclusions, the results of which are inconclusive for eczema [3], but essential for food allergy, in an attempt to minimise accidental—and potentially fatal—reactions [2]. However, these exclusions may impact diet quality [4], nutrient intake [4,5,6] and nutrient demands [7].

Adolescence is a critical window for adequate calcium intake, due to intensive bone and muscle development, and the need to optimise peak bone mass [8]. Milk is the top food source of calcium for children aged 2–18 years [9]. In a typical western diet, cow’s milk is a common source of calcium. Yet, milk is the most commonly avoided food amongst those with eczema [10] and a common allergen amongst children [2]. Children with a history of cow’s milk allergy, but who subsequently develop tolerance, continue to exhibit altered eating behaviours and food preferences into late childhood and early adolescence [11,12]. This suggests that cow’s milk allergy has the potential to impact overall diet, dietary choices, and taste preferences, even after tolerance is achieved. It is unknown whether such behaviours and preferences exist amongst children with allergies to other foods or eczema. Moreover, it is unclear if these adolescents compensate for low cow’s milk intake by increasing their consumption of other high-calcium foods, which could include leafy greens, calcium-fortified orange juice, and multivitamin/mineral supplements. To this end, the aim of this study was to examine the quality and intake of calcium-rich foods amongst adolescents with or without eczema and/or food allergy in childhood. Calcium intake was defined based on the frequency and quality of calcium-rich foods consumed, including dairy products, leafy greens, multivitamins and calcium-fortified orange juice, based on a modified version of the Youth Healthy Eating Index [13]. In this index, low-fat milk and yoghurt consumption were scored twice as high as high-fat, high-sugar dairy. We hypothesized that calcium intake in adolescence would be lower amongst children with vs. without eczema and/or food allergy in childhood.

## 2. Materials and Methods

This study makes use of data from the Study of Allergy, Genes and the Environment (SAGE), a nested case-control study of the 1995 Manitoba, Canada birth cohort, which has been detailed elsewhere [14]. In brief, 723 children were followed biennially from ages 7–8 (childhood), to 12–14 years (adolescence), at which time 489 (67.6%) children remained in the study. Of those who participated in the adolescent visit, information on eczema and food allergy in childhood, and dietary data in adolescence (detailed below) were available for 468 (95.9%) participants in the study. This constituted our study population. Participants for whom the exposure and outcome data were not available were excluded from the analysis.

In childhood, parents (predominantly mothers) self-declared ethnicity, which we classified as Caucasian; visible minority; or, Indigenous. Household income was also reported, which we dichotomized as being higher or lower than the provincial median at the childhood visit [15]. Mothers also reported no or yes to questions on completing any post-secondary education, smoking ever, any history of allergic disease, or ever breastfeeding the index child. Owing to wide intra-provincial variation in food availability and prices [16], we also considered region of residence within Manitoba (urban, southern rural, northern). As dietary intake may also be influenced by body mass index (BMI), we considered BMI in childhood and adolescence as appropriate for the various analyses.

### 2.1. Food Allergy and Eczema

Children were assessed by a pediatric allergist for eczema and food allergy as part of their clinical assessments in childhood and adolescence. From these questions, we created the following definitions:Eczema in childhood: Eczema diagnosis within the 12 months before the childhood visit.Eczema in adolescence: Eczema diagnosis within the 12 months before the adolescent visit.Persistent eczema: Eczema in childhood, and in adolescence.Transient eczema: Eczema in childhood, but not in adolescence.Eczema never: No eczema reported, to adolescence.Food allergy in childhood: Food allergy diagnosis within the 12 months before the childhood visit.Food allergy in adolescence: Food allergy diagnosis within the 12 months before the adolescent visit.Persistent food allergy: Food allergy in childhood, and in adolescence.Transient food allergy: Food allergy in childhood, but not in adolescence.Food allergy never: No food allergy reported, to adolescence.

### 2.2. Dietary Data

When adolescents were aged 12–14 years, they self-reported dietary data, with parental assistance as necessary, using a 39-item food frequency questionnaire (FFQ), adapted from the Nurses’ Health Study (NHS) [17]. The adapted FFQ has been described in a previous report from our group [18]. In brief, we excluded coffee and tea, alcoholic beverages, liver, and hard and cottage cheese. The primary outcome measure in this study was calcium intake in adolescence. Calcium intake is defined as the frequency and quality of calcium products consumed.

### 2.3. Calcium Intake: Quality

The calcium intake score includes the quality of dairy product consumption, specifically, the consumption of skim milk/2% milk, homogenized/whole milk, yoghurt, and ice cream/frozen yoghurt. It also scores low-fat milk and yoghurt twice as high as ice cream/frozen yoghurt and whole milk due to the high-fat and/or high-sugar content of these products and the availability of lower-fat milk alternatives. The calcium intake score was classified based on an adapted version of the Youth Healthy Eating Index (YHEI) [13]. The YHEI is derived from the United States’ Department of Agriculture’s Dietary Guidelines for Americans [19]. These guidelines are similar to those outlined in the then-current Canada’s Food Guide to Healthy Eating [20]. The FFQ used in this study was not originally designed to assess calcium intake using the YHEI. Thus, some adaptations to the scoring system were necessary. The minimum and maximum possible scores were 0 and 10, respectively.

### 2.4. Calcium Intake: Frequency

Specific to the present study, the frequency of cow’s milk, ice cream/frozen yoghurt and leafy greens intake were reported on an 8-point scale from 6+ times per day to almost never. Consideration was also given to multivitamin/mineral supplements, for which possible answers included often (once a day or more), sometimes (once a week or more), rarely (less than once a week), or never. From these answers, we created a dichotomous intake pattern of at least once weekly vs. less than once weekly. We also included food items that were not part of the NHS but were thought to be relevant. For example, calcium-fortified orange juice was introduced into the Manitoba food market around the time of adolescent visit. Thus, we added the question, “Does your child drink calcium-fortified orange juice?” Possible answers were no or yes. We acknowledge that soy/tofu may be an alternative source of calcium. However, we excluded soy/tofu from our analysis, as 95% of participants reported “almost never” consuming these products.

### 2.5. Statistical Analysis

Descriptive statistics were used to describe the study population. Linear regression analyses were used to examine associations between eczema and food allergy in childhood, adolescence, and timing, and calcium intake in adolescence. Results are reported as coefficients (β) and 95th percent confidence intervals (95%CI). Binary logistic regression was used, to examine potential associations between allergy status and calcium-rich food intake. Results are presented as odds ratios (OR) with corresponding 95th percent confidence intervals (OR; 95%CI). Statistical significance was set at *p* < 0.05. Analyses were performed with Stata SE 15 (Stata Corp, College Station, TX, USA). For all regression analyses, fully adjusted models were adjusted for breastfeeding, maternal education, maternal smoking, region of residence and child’s body mass index at the corresponding age. Sensitivity analyses using parent-reported sex of the child were performed to consider possible differences between boys and girls. The University of Manitoba Health Research Ethics Board approved this study (HS14742 (H2002:078)). Parents provided informed written consent and children provided assent.

## 3. Results

In total, 468 adolescents (64.7% retention from baseline) were included in this study, of whom 62 (13.3%) had eczema only in childhood, 25 (5.3%) with food allergy only, and 26 (5.6%) with eczema and food allergy. Boys and girls were equally represented, and a similar number of children were from households below and above the provincial median income (Table 1). The majority of children were Caucasian and lived in urban and southern rural communities. Most mothers had post-secondary education and less than half reported smoking ever. Approximately 60% of mothers had at least one allergic condition. But, as expected, these rates were higher amongst mothers whose children had both eczema and food allergy. Participants who had never had eczema or food allergy tended to have the highest calcium intake scores in adolescence, whereas the lowest scores were noted for those with eczema in childhood (4.20 ± 1.68, and 3.66 ± 1.91, respectively).

The majority of adolescents consumed milk and leafy greens at least once weekly. In contrast, weekly consumption patterns of ice cream/frozen yoghurt were varied. Only one-third of adolescents had multivitamin/mineral supplements weekly or consumed calcium-fortified orange juice (Figure 1). The frequency of high-calcium food consumption for all adolescents overall is shown in Table A1.

Compared to children without eczema, those with eczema, but without food allergy, had poorer calcium intake, in terms of both quality (β −0.44; 95%CI −0.96; 0.00) and frequency in adolescence (Table 2). Girls, but not boys, with eczema in childhood had poorer calcium intake in adolescence than girls without eczema (β −0.84; 95%CI −1.60; −0.08). In adolescence, those with eczema continued to have worse calcium intake, although this difference did not quite reach statistical significance (β −0.60; 95%CI −1.25; 0.04; *p* = 0.06). Similar but non-statistically significant patterns were found for food allergy in childhood and adolescence, and for calcium intake.

With consideration of intake of calcium-rich foods, compared to adolescents with neither food allergy nor eczema, those with eczema only were significantly less likely to consume ice cream or frozen yoghurt at least once weekly (OR 0.44; 95%CI 0.25; 0.78; Table A2). Similar but non-significant patterns were identified for skim milk (OR 0.59; 95%CI 0.29; 1.19) and leafy greens (OR 0.60; 95%CI 0.26; 1.41) intake. As few adolescents reported consuming homogenized (3.25% or “whole”) milk, we were not able to perform an analysis on this type of milk. Amongst children with food allergy only, or both eczema and food allergy, skim milk consumption followed a similar pattern to that seen for eczema, but again did not reach statistical significance. When stratified by sex, results were comparable to those for both genders combined (results not shown). The exception was for ice cream and frozen yoghurt intake amongst children with eczema. Whereas no differences in intake were found for boys, girls with eczema in childhood had a significantly lower odds of ice cream and frozen yoghurt intake in adolescence compared to girls with neither food allergy nor eczema in childhood (OR 0.34; 95%CI 0.16; 0.72).

Amongst those with eczema and/or food allergy in childhood, no significant difference in calcium intake was identified between adolescents with transient vs. persistent disease (Table 3). This observation was identified even for adolescents with eczema, despite lower dairy quality in childhood. No differences in the intake of any high-calcium foods—including milk—were found between those with transient vs. persistent disease. This included no difference in ice cream/frozen yoghurt intake in boys or girls amongst those with transient vs. persistent eczema.

## 4. Discussion

In this study of Manitobans born in 1995, seen initially in childhood, and followed prospectively to adolescence, eczema was associated with significantly lower calcium intake in adolescence. Similar but non-significant patterns of calcium intake were found for food allergy. Interestingly, for both eczema and food allergy, calcium intakes were comparable between those with persistent vs. transient disease. Girls, but not boys, with eczema had significantly lower intakes of ice cream and frozen yoghurt, a difference which was attenuated with consideration to transient vs. persistent disease when stratified by sex. The reasons for these gender-specific differences are unclear. Nonetheless, our collective findings suggest that dietary habits formed in childhood do not change, even when a child “outgrows” his/her eczema or food allergy.

Although the then-current Canada Food Guide advised selecting lower-fat milk products more often [20], ice cream/frozen yoghurt were nonetheless consumed by most adolescents on a weekly basis. This observation is important for two reasons. First, it provides evidence that these adolescents tolerate dairy products. And second, it supports a growing body of literature that adolescents are regularly choosing high fat, high sugar dairy products [21,22]. Elsewhere, such products have been linked with an increased risk of other chronic diseases, including risk factors for cardiometabolic disease [23,24]. For these reasons, reduced consumption of these foods has been a focus on the Canada Food Guide to Healthy Eating for many years. Although the majority of adolescents in our study reported drinking skim milk, rather than homogenized (3.25%) milk, most still consumed ice cream or frozen yoghurt as well. This contrasts with reports from other high-income countries, where dietary intake of dairy products amongst adolescents is low [9,24,25]. At a population level, this is encouraging. However, caution in interpretation is warranted, given the significantly lower consumption amongst children with eczema. Focused efforts, informed by adolescent participant engagement, should target ways in which to alter consumption patterns to improve milk intake and align consumption with the 2019 Canada Food Guide. This updated version of the Canada Food Guide has shifted focus from four food groups (fruits/vegetables, grains, meats, and dairy), to a plate model, emphasizing plenty of fruits and vegetables, whole grain carbohydrates, and plant-based protein [26]. This new guide removes dairy as its own food group and instead collapses it into the protein group. The 2019 Canada Food Guide does not provide any specific recommendations for calcium intake. Due to these new guidelines, it is possible that calcium intake among today’s children will decline. Indeed, there are already data to support that calcium intake is decreasing amongst Canadian adolescents. From 2004 to 2015, Canadian dietary patterns shifted toward significantly more dairy products, but less fluid milk, as well as lower consumption of vegetables [27].

In our study, eczema and food allergy in childhood were associated with lower calcium intake in adolescence. Although some evidence supports that milk consumption has been associated with lower frequencies of allergic disease, including eczema and sensitization to foods, these observations are largely restricted to fresh or raw milk [28,29,30], and have not been replicated in pasteurized milk or milk products. Rather, dairy products are often avoided by those with allergic diseases, including eczema and food allergy.

Many parents will practice avoidance of certain foods, and in particular, cow’s milk, in their children with eczema [10]. Previously, research from Hong Kong supports that children with eczema consume less milk (*p* < 0.06) compared to children without eczema [30]. This observation is noteworthy given that daily milk intake in that region is already uncommon [31], yet differences were still noted by eczema status. This suggests that physician–parent communication in the management of eczema may be inadequate, as current practice guidelines indicate that dietary restrictions in patients with eczema only are not beneficial [3]. Children and adults with eczema commonly avoid dairy, in particular, cow’s milk, to reduce symptoms [10]. Concerningly, dairy avoidance is done without consulting a dietitian [10]. With consideration of food allergy, British researchers reported that infants following a cow’s milk exclusion diet had significantly different taste preferences [12] and avoidance of certain foods [11] at age 11.5 years, than their peers who had been on an unrestricted diet in infancy. These studies did not compare the presence or absence of eczema or food allergy in early life. However, coupled with our findings, they suggest that avoidance of foods in infancy and childhood alters dietary preferences. Our findings extend this British study, as they suggest that any childhood food allergy may influence food choice in adolescence. Yet, adolescence is a critical window for calcium deposition. In a recent systematic review, adolescents who followed a typical diet but supplemented by dairy products had a significantly increased bone mineral content [32]. Moreover, in Canada, milk, but not other calcium-rich foods, are fortified with Vitamin D. Our data did not permit consideration to Vitamin D deficiency. Although cow’s milk allergy, in particular, appears to have a lasting effect on diet and feeding [33], our study supports that eczema also has long-term impacts on food choice and dietary patterns. We also provide evidence that querying “dairy” or “calcium” without consideration to the foods included in this category may inadequately capture adolescents’ calcium intake and fail to capture nutritionally and statistically significant differences between adolescents with and without food allergy.

We also acknowledge the small sample sizes of adolescents with consideration to persistent vs. transient eczema or food allergy. In addition, the validated YHEI [13] was adapted based on available data, thereby necessitating some exclusions. Of relevance to calcium intake, cottage and hard cheeses were not included in the FFQ to prevent respondent fatigue. These items, if consumed regularly, could have contributed to overall calcium intake but were not considered in our analysis. In addition, as with all self-reported dietary data, FFQ have intrinsic limitations. Such limitations range from concerns with variations in nutrient composition and portion sizes of the reported foods. These latter three issues may have been further compounded by the fact that adolescents were asked to complete the FFQ with assistance as needed from their parents, which improves validity [34]. Finally, we acknowledge the limitations of the FFQ. It was only administered at a single time point. If the FFQ had been administered in both adolescence and childhood, we could have completed a cross-sectional analysis between eczema and dairy quality in childhood, making our results more complete. We also acknowledge that cheese intake was not queried in our FFQ.

Strengths of our study include a province-wide study population, in which diverse ethnicities and socio-economic statuses are represented, but which also approximate the demographic distribution of our province [35], thereby increasing the generalizability of our findings. Both eczema and food allergy were based on a specialist assessment, thus minimizing the likelihood of disease misclassification. This is a significant strength compared to studies from other birth cohorts, in which eczema and food allergy were based on parental report [36,37,38]. Our FFQ included numerous types of high-calcium foods, including some, such as calcium-fortified orange juice, which were recent introductions to the food market at the time our FFQ was developed. Finally, we highlight that we relied on adolescent-reported dietary intake, given that the agreement between parent-report and adolescent intake is usually poor [39].

## 5. Conclusions

In conclusion, compared to their peers with neither eczema nor food allergy, the majority of Manitoba adolescents with eczema only in childhood have poorer calcium intake and lower intakes of ice cream and frozen yoghurt, which are higher in calories, but which have comparable calcium bioavailability to milk [40]. These differences persist through adolescence whether the child has transient or persistent disease. Dietary education monitoring, with attention to both foods and nutrients, is important for adolescents with eczema or food allergy. Likewise, families whose children and adolescents with eczema and/or food allergy avoid dairy should be encouraged to make substitutions with other calcium-rich foods.

## Figures and Tables

**Figure 1 nutrients-11-03039-f001:**
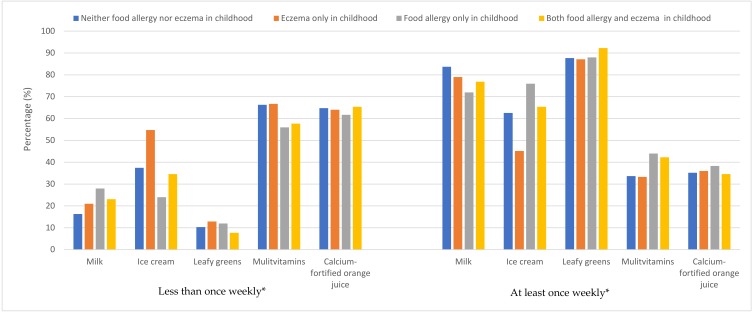
Intake patterns of calcium-rich foods in adolescence, amongst children with neither food allergy nor eczema, or one or both conditions. * Calcium-fortified orange juice consumption defined as no vs. yes.

**Table 1 nutrients-11-03039-t001:** Demographic and characteristics of the child and family (*N* = 468).

	Neither Food Allergy nor Eczema in Childhood (*N* = 355)		Eczema only in Childhood (*N* = 62)		Food Allergy only in Childhood (*N* = 25)		Both Food Allergy and Eczema in Childhood (*N* = 26)	
Family Demographics	*n*	%	*n*	%	*n*	%	*n*	%
Sex								
Male	200	56.3	31	50.0	19	76.0	13	50.0
Female	155	43.7	31	50.0	6	24.0	13	50.0
Household income								
$60,000 or less	150	45.9	26	46.4	12	48.0	11	55.00
$60,001 or more	177	54.1	30	53.6	13	52.0	9	45.00
Characteristics of Child								
Breastfed ever	302	85.6	58	93.6	23	92.0	24	92.3
Body mass index (kg/m^2^) in childhood	18.8 ± 3.7		17.9 ± 3.0		17.0 ± 2.2		18.2 ± 4.1	
Body mass index (kg/m^2^) in adolescence	20.6 ± 4.7		20.6 ± 4.0		19.2 ± 3.7		20.6 ± 5.0	
Calcium intake scores in adolescence *	4.20 ± 1.68		3.99 ± 1.55		3.66 ± 1.91		3.89 ± 2.46	
Characteristics of Mom								
Post-secondary education	299	88.5	53	89.8	25	100	21	95.5
Ever smoker	149	42.6	19	30.7	11	44.0	12	50.0
Region of residence								
Urban	186	52.4	43	69.4	15	60.0	18	69.2
Southern rural	139	39.1	18	29.0	10	40.0	7	26.9
Northern	30	8.5	1	1.6	0	0.0	1	3.9
At least 1 allergic disease	221	63.1	45	72.6	14	56.0	20	83.3

* Based the modified YHEI [18], with minimum and maximum possible scores of 0 and 10, respectively.

**Table 2 nutrients-11-03039-t002:** Calcium quality amongst adolescents with vs. without food allergy and/or eczema in childhood or adolescence.

	Unadjusted				Model 1 *		Model 2 ^†^	
I: Childhood	*n*	%	β	95% CI	β	95% CI	β	95% CI
Neither eczema nor food allergy	355	75.9	Ref		Ref		Ref	
Food allergy	25	5.3	−0.00	0.70; 0.69	−0.0	−0.70; 0.69	−0.05	−0.75; 0.64
Eczema	62	13.2	−0.48	−0.97; 0.01	−0.48	−0.97; 0.09	−0.44	−0.96; 0.00
Eczema and food allergy	26	5.6	−0.32	−1.00; 0.36	−0.32	−1.18; 0.35	−0.32	−1.00; 0.36
II: Adolescence								
Neither eczema nor food allergy	399	85.4	Ref		Ref		Ref	
Food allergy	26	5.6	−0.22	−0.91; 0.47	−0.18	−0.91; 0.54	−0.13	−0.87; 0.62
Eczema	34	7.3	−0.55	−1.17; 0.08	−0.60	−1.24; 0.04	−0.60	−1.25; 0.04
Eczema and food allergy	8	1.7	−0.31	−1.51; 0.89	−0.23	−1.52; 1.06	−0.14	−1.44; 1.15

* Adjusted for breastfeeding and maternal education; ^†^ Adjusted for breastfeeding ever, maternal education, maternal smoking, region of residence and, maternal smoking, and child’s BMI in childhood or adolescence as appropriate.

**Table 3 nutrients-11-03039-t003:** Calcium quality amongst adolescents persistent vs. transient food allergy or eczema.

	Unadjusted				Model 1 ^†^		Model 2 ^‡^	
Eczema (*N* = 60*)	*n*	%	β	95% CI	β	95% CI	β	95% CI
Transient	43	71.7	Ref		Ref		Ref	
Persistent	17	28.3	−0.94	−2.03; 0.14	−1.07	−2.16; 0.04	−0.93	−2.11; 0.22
Food allergy (*N* = 25)								
Transient	13	52.0	Ref		Ref		Ref	
Persistent	12	48.0	−0.12	−2.01; 1.77	−0.96	−2.94; 1.03	−0.75	−2.83; 1.33
Food allergy & eczema (*N* = 26)								
Transient	11	44.0	Ref		Ref		Ref	
Persistent	14	56.0	−0.44	−1.70; 0.81	−0.48	−2.13; 1.16	−0.75	−2.63; 1.13

* Two participants lost to follow up between the childhood and adolescent visits; ^†^ Adjusted for breastfeeding and maternal education; ^‡^ Adjusted for breastfeeding ever, maternal education, maternal smoking, region of residence and, maternal smoking, and child’s BMI in adolescence.

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
