# Peer review of "Calcium Intake in Children with Eczema and/or Food Allergy: A Prospective Cohort Study"

_nutrients, 2019, doi:10.3390/nu11123039_

Round 1

Reviewer 1 Report

This paper is well-written and addresses a unique topic that has drawn a great deal of attention lately from the scientific and consumer communities. The findings of this paper are timely and important. However, the presentation of findings requires greater context upfront from the authors and the framing of the discussion needs work as well.

A few overarching comments:

“dairy quality” is used throughout the paper. While it is defined on lines 90-101, its use in the abstract without definition/context makes it difficult to understand what it means. Even after reading the definition, it remains unclear what exactly the authors mean by “quality”- at different points, it seems to mean “nutrient-density” and at others, it seems to mean “calcium content.” Please provide clarification throughout the paper or consider alternative terminology. Allergy and eczema seem to become conflated at some points in this paper. Recommend carefully separating the two, especially in the introduction section and in the discussion section- with implications, which will necessarily be different for the two conditions.

Introduction

Line 33: is this prevalence of eczema specific to Canada? Or is it global?

38-9; are there other reasons that adolescents would need to consume adequate amounts of dairy foods? In the U.S., they are also a major source of several other nutrients for children and adolescents- true in Canada as well?

47-8: how were these foods identified as alternative sources of calcium? What about fortified soy beverage? Reference needed. Bioavailability of calcium varies considerably between them and would be worth noting.

49: what does quality of calcium-rich foods (‘dairy’) mean? Especially after you have just defined calcium-rich foods to include a longer list?

50-52: more upfront explanation of ‘dairy quality’ needed- at first glance, appears to be a reference to the calcium content

Materials and Methods

63-4: The parent in the study self-identified their ethnicity- why did you categorize them? Were there differences between your categories and the original answers that could have affected your groups?

92: is dairy quality about the frequency as well? This is unclear, since the very next section addresses dairy intake, which would seem to reflect frequency. Please clarify.

Why is cheese omitted from this section? You address later but may want to include here as well- it is a noticeable omission. How were these dairy foods selected? How was 2% milk categorized?

95: Whole milk is not a ‘high-sugar’ food.

102: why only these items (milk, ice cream/frozen yogurt and leafy greens)?

Table 1- ethnicity is missing- is there a reason?

Results

158: in the U.S. “homogenized” refers to even distribution of fat droplets throughout milk but it seems here to refer to the fat content of the milk. Please clarify.

167-8: is this sentence meant to say “dairy quality”? It is the first time “diet quality” has appeared in this paper.

Discussion

181-3: Do you have an indication that the diets of these adolescents have remained largely unchanged, given that you do not have intake data from childhood? Please provide a justification for being able to make this remark. It is possible that a child “outgrew” an allergy, started drinking milk, then stopped in adolescence- without evidence otherwise, this statement is extrapolation.

187-8: Please provide a reference for this statement about adolescents primarily choosing high-fat/high-sugar dairy foods more often- where is the growing body of evidence?

189: Source 19 is on milk, not ice cream, and these two references are not adequate to make a claim that high-fat dairy is linked with cardiometabolic disease. Body of literature as a whole suggests that the links are rather neutral (between consumption of whole-fat dairy and cardiovascular disease).

As a consideration, if adolescents are consuming ice cream on a weekly basis, is this cause for concern? Is there evidence showing that weekly dietary ‘treats’ are indicative of longer term health problems? Or is there other evidence about the quality of the diets of Canadian adolescents as a whole that suggests that they are rather too high in fat and sugar? Calling out one isolated component of the diet (weekly ice cream consumption), this argument falls flat.

Source 21 is on milk, there are other sources that would be a better fit here (reflective of dairy intake as a whole among children). See Fulgoni paper on nutrients of concern from 2018.

200-202: what do the new Canadian food guides mean for calcium intake among adolescents?

203-207: Are there confounding variables in these studies of raw milk? i.e. seem to be conducted in children who grew up on farms. Are there other factors, besides milk intake, that could have contributed to lower rates of allergy/eczema? Recent papers on food allergy and atopic dermatitis in AJCN may be helpful here. Any evidence on why dairy foods are avoided by those with eczema? Would be useful to touch on that as well, not just allergy.

217: Please change “infancy” to “early childhood”- infants should never been given cow’s milk, due to risk for internal bleeding when provided to children under 1 year of age.

232-233: respondent fatigue seems an unconvincing argument here, especially given that these foods are sources of calcium, ostensibly an important part of this study given the upfront context. Why look at ice cream rather than hard cheese intake?

242-243: Results section indicates that this sample was not terribly diverse in terms of ethnicity (lines 129-34)- please clarify. 

253: source for stating that ice cream has lower calcium bioavailability than milk? Likely has lower amounts but it’s not clear why the bioavailability would differ among dairy products. It would differ among plant products, fortified foods, supplements, and dairy foods, though.

255-7: This conclusion could be fleshed out more- what does this say about the need for nutrition education for children with eczema and their families? Or the importance of dietary variety? And not restricting foods unless necessary?

Author Response

Reviewer 1.

____________________________________________________________________________________

This paper is well-written and addresses a unique topic that has drawn a great deal of attention lately from the scientific and consumer communities. The findings of this paper are timely and important. However, the presentation of findings requires greater context upfront from the authors and the framing of the discussion needs work as well.

Response:

Thank you for this comment. We are grateful that Reviewer 1 found this paper to be timely and important. Below, we have provided point-by-point responses to your careful comments, which we appreciate and believe improved the manuscript.

A few overarching comments:

Comment:

“dairy quality” is used throughout the paper. While it is defined on lines 90-101, its use in the abstract without definition/context makes it difficult to understand what it means. Even after reading the definition, it remains unclear what exactly the authors mean by “quality”- at different points, it seems to mean “nutrient-density” and at others, it seems to mean “calcium content.” Please provide clarification throughout the paper or consider alternative terminology. Allergy and eczema seem to become conflated at some points in this paper. Recommend carefully separating the two, especially in the introduction section and in the discussion section- with implications, which will necessarily be different for the two conditions.

Response:

Thank you for this comment. We have now revised “dairy quality” to “calcium intake” and have updated the definition to reflect the comprehensive score composition. Specifically, calcium intake has been defined as: the frequency and quality of calcium products consumed. (Please see Abstract, Lines 17-18; Methods, Lines 97-98).

Similarly, we have now revised allergic disease to reflect eczema and/or food allergy, as appropriate, in the Discussion (Please see Discussion, Lines 191-2, 214-5; 262).

Introduction

Comment:

Line 33: is this prevalence of eczema specific to Canada? Or is it global?

Response:

This prevalence estimate is based on a systematic review and meta-analysis of longitudinal studies, which included data from 6 different countries. We have added the term “global” to this sentence to provide the reader with greater ability to contextualize the estimated prevalence of eczema reported herein. (Please see Introduction, Line 34).

Comment:

38-9; are there other reasons that adolescents would need to consume adequate amounts of dairy foods? In the U.S., they are also a major source of several other nutrients for children and adolescents- true in Canada as well?

Response:

Whereas we acknowledge there are other calcium-foods that adolescents, milk remains the major source of calcium for children aged 2-18 years. (Reference: O’Neill CE et al. Nutrients. 2018).

Comment:

47-8: how were these foods identified as alternative sources of calcium? What about fortified soy beverage? Reference needed. Bioavailability of calcium varies considerably between them and would be worth noting.

Response:

These foods were identified as alternative sources of calcium based on the Food Frequency Questionnaire from the Nurses’ Health Study (Reference: Willett WC. Nutritional Epidemiology. 1990).

Whereas Manitoba is now a major producer of soybeans, these products were not widely available on the market, and thus likely not commonly consumed, at the time the nutrient data were collected (in 2007-10). As described by Manitoba Pulse, the prairie provinces, including Manitoba, have seen a 4% annual increase in soybean production (Manitoba Pulse, 2015). This is reinforced by our data, in which we found that 94.7% (430/454) of participants “almost never” consumed soy/tofu. This has been added to the Methods, Section 2.4 (Please see Lines 120-122).

Comment:

49: what does quality of calcium-rich foods (‘dairy’) mean? Especially after you have just defined calcium-rich foods to include a longer list?

Response:

Thank you for this comment. We have deleted (‘dairy’), such that the sentence now reads: To this end, the aim of this study was to examine the quality and intake of calcium-rich foods (‘dairy’) amongst adolescents with or without eczema and/or food allergy in childhood. (Please see Lines 53-55).

Comment:

50-52: more upfront explanation of ‘dairy quality’ needed- at first glance, appears to be a reference to the calcium content

Response:

Thank you for this comment. As noted above, we have now revised “dairy quality” to “calcium intake” and have updated the definition to reflect the comprehensive score composition. Specifically, calcium intake has been defined as: the frequency and quality of calcium products consumed. (Please see Abstract, Lines 17-18; Methods, Lines 100-101).

Materials and Methods

Comment:

63-4: The parent in the study self-identified their ethnicity- why did you categorize them? Were there differences between your categories and the original answers that could have affected your groups?

Response:

We categorized parent-reported ethnicity to protect the identities of some participants who identified with a particular visible minority group (e.g. from individual Asian and South Asian ethnicities). The resulting categorization did not affect the groups.

Comment:

92: is dairy quality about the frequency as well? This is unclear, since the very next section addresses dairy intake, which would seem to reflect frequency. Please clarify.

Response:

As noted above, we have now revised “dairy quality” to “calcium intake” and have updated the definition to reflect the comprehensive score composition. Specifically, calcium intake has been defined as: the frequency and quality of calcium products consumed. (Please see Abstract, Lines 17-18; Methods, Lines 100-101).

Comment:

Why is cheese omitted from this section? You address later but may want to include here as well- it is a noticeable omission. How were these dairy foods selected? How was 2% milk categorized?

Response:

These foods were identified as alternative sources of calcium based on the Food Frequency Questionnaire (FFQ) from the Nurses’ Health Study (Reference: Willett WC. Nutritional Epidemiology. 1990). Our FFQ did not include cheese. Whereas this could be regarded as a limitation of the study, we highlight that the original FFQ only included two questions on cheese (in the forms of cottage cheese, and hard cheese; Reference: Willett WC. Nutritional Epidemiology. 1990).

Comment:

95: Whole milk is not a ‘high-sugar’ food.

Response:

We agree. The sentence in the original submission read: It also scores low-fat milk and yoghurt twice as high as ice cream/frozen yoghurt and whole milk due to the high-fat, high-sugar content of these products and the availability of lower-fat milk alternatives.

We have revised this sentence to read: It also scores low-fat milk and yoghurt twice as high as ice cream/frozen yoghurt and whole milk due to the high-fat and/or high-sugar content of these products and the availability of lower-fat milk alternatives (Please see Line 103).

Comment:

102: why only these items (milk, ice cream/frozen yogurt and leafy greens)?

Response:

This score was based on an adaptation from the Youth Healthy Eating Index. No scoring system was available for multivitamins or for calcium-fortified orange juice (Reference: Feskanich et al. J Am Diet Assoc. 2004).

Comment:

Table 1- ethnicity is missing- is there a reason?

Response:

Parents were provided the opportunity to self-identify with a particular ethnicity. However, they also had the right to decline to answer this (and any other) question. We note that ethnicity data were available for 99.4% (719/723) of the overall cohort. Thus, the 4 missing data points are not likely to have impacted the results.

Results

Comment:

158: in the U.S. “homogenized” refers to even distribution of fat droplets throughout milk but it seems here to refer to the fat content of the milk. Please clarify.

Response:

In Canada, milk with a 3.25% milk fat content is referred to as homogenized milk.

Comment:

167-8: is this sentence meant to say “dairy quality”? It is the first time “diet quality” has appeared in this paper.

Response:

Thank you for this comment. We are unable to see any mention of “diet quality” in, or around Lines 167-8. Please advise if you are referring to “diet quality” in Line 107. If so, this has now been updated to read “calcium intake.”  (Please see Line 108).

Discussion

Comment:

181-3: Do you have an indication that the diets of these adolescents have remained largely unchanged, given that you do not have intake data from childhood? Please provide a justification for being able to make this remark. It is possible that a child “outgrew” an allergy, started drinking milk, then stopped in adolescence- without evidence otherwise, this statement is extrapolation.

Response:

Data from other large cohorts provide evidence that dietary patterns remain stable in childhood. For example, ALSPAC (Avon Longitudinal Study of Pregnancy and Childhood) researchers reported that dietary patterns were “virtually identical” from mid-childhood (Reference: Northstone et al. Br J Nutr. 2008).

Parents completed a very brief FFQ on behalf of their children at the childhood visit. Of the participants for whom milk intake data are available in both childhood and adolescence, 86% (365/426) reported at least once weekly milk consumption at both time points. Although we lack data in childhood on the frequency of intake of other calcium-rich foods, this evidence provides us with additional confidence that milk intake – as a proxy for calcium-intake – remains largely unchanged from childhood to adolescence.

Comment:

187-8: Please provide a reference for this statement about adolescents primarily choosing high-fat/high-sugar dairy foods more often- where is the growing body of evidence?

Response:

We have now added references to support this sentence (References: Rosinger et al, 2017; Poti et al, 2014)

Comment:

189: Source 19 is on milk, not ice cream, and these two references are not adequate to make a claim that high-fat dairy is linked with cardiometabolic disease. Body of literature as a whole suggests that the links are rather neutral (between consumption of whole-fat dairy and cardiovascular disease).

Response:

We have revised this sentence to read: Elsewhere, such products have been linked with an increased risk of other chronic diseases, including risk factors for cardiometabolic disease. Reference 20 (now Reference 24) has been updated to Johansson, I.; Nilsson, L.M.; Esberg, A.; Jansson, J.; Winkvist, A. Dairy intake revisted – associations between dairy intake and lifestyle related cardio-metabolic risk factors in a high milk consuming population. Nutr. J. 2018, 17(110); DOI: 10.1186/s12937-018-0418-y.

Comment:

As a consideration, if adolescents are consuming ice cream on a weekly basis, is this cause for concern? Is there evidence showing that weekly dietary ‘treats’ are indicative of longer term health problems? Or is there other evidence about the quality of the diets of Canadian adolescents as a whole that suggests that they are rather too high in fat and sugar? Calling out one isolated component of the diet (weekly ice cream consumption), this argument falls flat.

Response:

A recent publication, in Nutrients, based on Canadian data collected in 2004 and 2015, describes how Canadian dietary patterns have shifted toward significantly dairy products, but less fluid milk. The authors of this paper write: “the declines found in population-level mean intakes of milk and alternatives and fluid milk suggests that improvement in the adequacy of these nutrients of concern is unlikely and merits further attention.” Moreover, in the same population, there was a shift toward lower consumption of fruits and vegetables” particularly in adolescents. Taken together, this provides evidence that the dietary patterns of Canadian adolescents are shifting toward higher fat dairy/sugar dairy products and lower sources of other high calcium foods, including fluid milk and leafy greens. (Reference: Tugault-Lafleur CN et al. Nutrients. 2019).

Text describing the recent changes in dietary patterns has been added to the Discussion. (Please see Lines 221-223).

Comment:

Source 21 is on milk, there are other sources that would be a better fit here (reflective of dairy intake as a whole among children). See Fulgoni paper on nutrients of concern from 2018.

Response:

Thank you. We have now changed Reference 21 from Maillot et al. to:

O’Neil, C.E.; Niklas, T.A.; Fulgoni, V.L. 3rd. Food sources of energy and nutrients of public health concern and nutrients to limit wth a focus on milk and other diary foods in children 2 to 18 years of age: National Health and Nutrition Examination Survey, 2011-2014. Nutrients. 2018, 10(8), pii: E1050. doi: 10.3390/nu10081050.

Comment:

200-202: what do the new Canadian food guides mean for calcium intake among adolescents?

Response:

The following text has been added to the Discussion, in the paragraph on the Canada Food Guide:

The 2019 Canada Food Guide does not provide any specific recommendations for calcium intake. (Please see Lines 218-219).

Comment:

203-207: Are there confounding variables in these studies of raw milk? i.e. seem to be conducted in children who grew up on farms. Are there other factors, besides milk intake, that could have contributed to lower rates of allergy/eczema? Recent papers on food allergy and atopic dermatitis in AJCN may be helpful here. Any evidence on why dairy foods are avoided by those with eczema? Would be useful to touch on that as well, not just allergy.

Response:

These studies were based on children who lived in rural areas, but not exclusively on farms. In Canada, it is illegal to sell raw milk, an act which is a federal crime and punishable by up to 2 years in prison (Reference: Selick, 2018).

There are other risk factors that have been linked to allergic disease, including eczema and asthma. However, many of these risk factors, particularly environmental risk factors, are controlled for in the studies cited in References 24-25.

We have added the following two sentences to the Discussion, in the paragraph on avoidance. (Please see Lines 236-238).

Children and adults with eczema commonly avoid dairy, in particular, cow’s milk, to reduce symptoms [9]. Concerningly, dairy avoidance is done without consulting a dietitian [9].

Comment:

217: Please change “infancy” to “early childhood”- infants should never been given cow’s milk, due to risk for internal bleeding when provided to children under 1 year of age.

Response:

Thank you for this interesting comment. This is not advice that is given to Canadian parents (References: Canadian Paediatric Society Caring for Kids; Dietitians of Canada). As such, we have not changed the language in this sentence.

Comment:

232-233: respondent fatigue seems an unconvincing argument here, especially given that these foods are sources of calcium, ostensibly an important part of this study given the upfront context. Why look at ice cream rather than hard cheese intake?

Response:

It is indeed unfortunate that our FFQ did not query cheese consumption. This decision was made in 2006 when adapting the Nurses’ Health Study FFQ to the present study. If we were to repeat the same study, or use the same FFQ, we would query cheese intake. To this end, we have modified the sentence on limitations of the FFQ. (Please see Lines 266-267).

Finally, we acknowledge the limitations of the FFQ. It was only administered at a single time point. If the FFQ had was administered in both adolescence and childhood we could have completed a cross-sectional analysis between eczema and dairy quality in childhood making our results more complete. We also acknowledge that cheese intake was not queried in our FFQ.

Comment:

242-243: Results section indicates that this sample was not terribly diverse in terms of ethnicity (lines 129-34)- please clarify. 

Response:

The population in our study very much paralleled the ethnic distribution of our province. In Manitoba, approximately 20% identify as visible minorities (Reference: Statistics Canada). In our study population, 26% identified as visible minorities. This has been clarified in the Discussion. (Please see Lines 269-270).

Comment:

253: source for stating that ice cream has lower calcium bioavailability than milk? Likely has lower amounts but it’s not clear why the bioavailability would differ among dairy products. It would differ among plant products, fortified foods, supplements, and dairy foods, though.

Response:

Thank you. We have adjusted the language and provided a reference that indicates comparable calcium bioavailability between these foods (Reference: van der Hee et al. J Am Diet Assoc. 2009; Please see Lines 280-281).

Comment:

255-7: This conclusion could be fleshed out more- what does this say about the need for nutrition education for children with eczema and their families? Or the importance of dietary variety? And not restricting foods unless necessary?

Response:

Thank you. We have added “education” to dietary monitoring (Please see Line 284). Similarly, we have added the following sentence: Likewise, families whose children and adolescents with eczema and/or food allergy avoid dairy should be encouraged to make substitutions with other calcium-rich foods. (Please see Line 285-286).

References for Response to Reviewer 1.

Johansson I, Nilsson LM, esberg A, Jansson J-H, Winkvist. Dairy intake revisted – associatios between dairy intake and lifestyle related cardio-metabolic risk factors in a high milk consuming population. Nutr. J. 2018, 17(110); DOI: 10.1186/s12937-018-0418-y.

Reviewer 2 Report

The authors analyze dairy intake during adolescence as it relates to prior or current food allergy and/or eczema. They report lower dairy quality in adolescence for those children with eczema in childhood. 

Clarity in the methods is needed.

How was missing data handled? Were participants with any missing data eliminated from all analyses?

The adaptations made to the YHEI must be described in the text. Based on the scoring, (1) what was the possible minimum score? (2) what was the possible maximum score?

In the results section, the average (or median) and standard deviation (or min/max) YHEI score for the participants should be given in table 1.

Has the rest of the FFQ data from this study been published elsewhere? If yes, that should be stated and those papers should be cited.

In table 3, why are there only 60 participants in the eczema group instead of 62?

I am confused by the data presented in table A1. The text states that the non-atopic individuals were the reference group, but in the table, the <1 time/week consumers are listed as the reference group. If the <1 time/week consumers are the reference group, then what is the outcome that is being reported? The text states that it is the consumption times per week, but then that can't be the reference group. As presented in table A1, some outcome is being compared between those who consume <1 time/week versus those who consume 1 or more times per week. Perhaps I am confused. Please explain.

Please provide a table or figure (can be in main text or supplemental) to report the 8-point scale results from the FFQ for the overall population of adolescents.

The conclusion that most adolescents consume ice cream/frozen yoghurt is not supported by the data provided in the current version of the manuscript. A majority, yes, but not most. Most, for me, means more than 90%, but in all four groups shown in figure 1, none reaches 80% of its members consuming ice cream/frozen yoghurt.

The conclusion in line 203-204 is not supported by the data. This statement is only supported for dairy quality in those with eczema during childhood.

Typographical errors:

Table 2

Eczema and food allergy row in I:Childhood. The unadjusted beta appears to be missing a negative sign in front of the number.

Author Response

Reviewer 2.

_____________________________________________________________________________________

Comment:

The authors analyze dairy intake during adolescence as it relates to prior or current food allergy and/or eczema. They report lower dairy quality in adolescence for those children with eczema in childhood. 

Response:

Thank you. Kindly note in the revised manuscript we now describe dairy quality as “calcium intake.”

Comment:

Clarity in the methods is needed. How was missing data handled? Were participants with any missing data eliminated from all analyses?

Response:

As noted in Lines 67-68, our study population consisted of participants for whom “ information on eczema and food allergy in childhood, and dietary data in adolescence (detailed below) were available for 468 (95.9%) of participants in the study.” Participants for whom the exposure and outcome data were not available were excluded from the analysis.

Comment:

The adaptations made to the YHEI must be described in the text. Based on the scoring, (1) what was the possible minimum score? (2) what was the possible maximum score?

Response:

Exclusions made to the YHEI have now been added, as noted in Line 96: In brief, we excluded coffee and tea, alcoholic beverages, liver, and hard and cottage cheese.

The minimum and maximum possible scores have now been added, as noted in Lines 11109-110: The minimum and maximum possible scores were 0 and 10, respectively.

Comment:

In the results section, the average (or median) and standard deviation (or min/max) YHEI score for the participants should be given in table 1.

Response:

Thank you for this comment. We like it very much indeed. The mean and standard deviation scores have been added to Table 1, and summarized in Lines 145-147.

Comment:

Has the rest of the FFQ data from this study been published elsewhere? If yes, that should be stated and those papers should be cited.

Response:

Thank you for this suggestion. Yes, the FFQ is published in its entirety in a previous report from our group (Reference: Protudjer et al. 2012)

Comment:

In table 3, why are there only 60 participants in the eczema group instead of 62?

Response:

Unfortunately, 2 participants with eczema (but not food allergy) were lost to follow up between the childhood and adolescent visits. This has been clarified in a footnote in Table 1.

Comment:

I am confused by the data presented in table A1. The text states that the non-atopic individuals were the reference group, but in the table, the <1 time/week consumers are listed as the reference group. If the <1 time/week consumers are the reference group, then what is the outcome that is being reported? The text states that it is the consumption times per week, but then that can't be the reference group. As presented in table A1, some outcome is being compared between those who consume <1 time/week versus those who consume 1 or more times per week. Perhaps I am confused. Please explain.

Response:

The binary logistic regression analyses presented in Table A1 (now Table A2) consider the association between non-atopic individuals with participants with food allergy only (Panel I), eczema only (Panel II) and both food allergy and eczema (Panel IIII), and frequency on consumption (<1 weekly, vs. weekly).

As the table is already quite long, we have not presented the cross-tabs for each group (i.e. each Panel), and those with non-atopic individuals.

Comment:

Please provide a table or figure (can be in main text or supplemental) to report the 8-point scale results from the FFQ for the overall population of adolescents.

Response:

This information is now presented in Table A1, with a reference to the Table in Line 154.

Comment:

The conclusion that most adolescents consume ice cream/frozen yoghurt is not supported by the data provided in the current version of the manuscript. A majority, yes, but not most. Most, for me, means more than 90%, but in all four groups shown in figure 1, none reaches 80% of its members consuming ice cream/frozen yoghurt.

Response:

We have added the phrase “the majority” to the conclusion as suggested (Lines 279).

Comment:

The conclusion in line 203-204 is not supported by the data. This statement is only supported for dairy quality in those with eczema during childhood.

Response:

We have deleted this sentence.

Typographical errors:

Comment: Table 2: Eczema and food allergy row in I:Childhood. The unadjusted beta appears to be missing a negative sign in front of the number.

Response:

Thank you. Yes, you are correct. The change has been made.

References for Reviewer 2

Protudjer JL, Sevenhuysen GP, Ramsey CD, Kozyrskyj AL, Becker AB. Low vegetable intake is associated with allergic asthma and moderate-to-severe airway hyperresponsiveness. Pediatr Pulmonol. 2012;47(12)1159-69.  doi: 10.1002/ppul.22576. 

Round 2

Reviewer 1 Report

The authors have thoroughly edited and revised this manuscript to align with previous feedback. 

Author Response

Thank you. We are are pleased that Reviewer 1 found the edits to be satisfactory.

Reviewer 2 Report

Thank you for addressing my concerns.

Two minor outstanding issues:

I believe that there needs to be a correction to the footnote in Table 1. I was under the impression that the score in table 1 is actually based on reference 18 (the modified YHEI) rather than the original YHEI reference.

I also ask that the authors mention in the abstract that cheese was not included in the assessment of calcium rich foods. For instance, "...examined calcium intake, defined as the frequency and quality of calcium products (with the exception of cheese) consumed, amongst Manitoba..."

Otherwise, I have no further comments on this manuscript.

Author Response

We are grateful that Reviewer 2 was satisfied with the edits. Below are point-by-point responses to the two additional comments.

Comment 1:

I believe that there needs to be a correction to the footnote in Table 1. I was under the impression that the score in table 1 is actually based on reference 18 (the modified YHEI) rather than the original YHEI reference.

Response to Comment 1:

Thank you for catching this. You are correct. The change has been made. (Please see Table 1, Footnote).

Comment 2:

I also ask that the authors mention in the abstract that cheese was not included in the assessment of calcium rich foods. For instance, "...examined calcium intake, defined as the frequency and quality of calcium products (with the exception of cheese) consumed, amongst Manitoba..."

Response to Comment 2:

We have incorporated this suggestion into the abstract. Please see Line 17.